# Rheumatic Immune-Related Adverse Events due to Immune Checkpoint Inhibitors—A 2023 Update

**DOI:** 10.3390/ijms24065643

**Published:** 2023-03-15

**Authors:** Quang Minh Dang, Ryu Watanabe, Mayu Shiomi, Kazuo Fukumoto, Tomomi W. Nobashi, Tadashi Okano, Shinsuke Yamada, Motomu Hashimoto

**Affiliations:** 1Department of Clinical Immunology, Osaka Metropolitan University Graduate School of Medicine, Osaka 565-0871, Japan; 2Preemptive Medicine and Lifestyle Related Disease Research Center, Kyoto University Hospital, Kyoto 606-8397, Japan; 3Department of Orthopaedic Surgery, Osaka Metropolitan University Graduate School of Medicine, Osaka 565-0871, Japan

**Keywords:** checkpoint inhibitor, immune-related adverse events, malignancy, rheumatoid arthritis

## Abstract

With the aging of the population, malignancies are becoming common complications in patients with rheumatoid arthritis (RA), particularly in elderly patients. Such malignancies often interfere with RA treatment. Among several therapeutic agents, immune checkpoint inhibitors (ICIs) which antagonize immunological brakes on T lymphocytes have emerged as a promising treatment option for a variety of malignancies. In parallel, evidence has accumulated that ICIs are associated with numerous immune-related adverse events (irAEs), such as hypophysitis, myocarditis, pneumonitis, and colitis. Moreover, ICIs not only exacerbate pre-existing autoimmune diseases, but also cause de novo rheumatic disease–like symptoms, such as arthritis, myositis, and vasculitis, which are currently termed rheumatic irAEs. Rheumatic irAEs differ from classical rheumatic diseases in multiple aspects, and treatment should be individualized based on the severity. Close collaboration with oncologists is critical for preventing irreversible organ damage. This review summarizes the current evidence regarding the mechanisms and management of rheumatic irAEs with focus on arthritis, myositis, and vasculitis. Based on these findings, potential therapeutic strategies against rheumatic irAEs are discussed.

## 1. Introduction

Rheumatoid arthritis (RA) is an autoimmune disorder associated with progressive joint destruction and reduced physical function [1]. Over the past few decades, treatment options, such as biological agents and molecularly targeted antirheumatic drugs, have dramatically expanded, making it a feasible therapeutic goal for many patients to achieve clinical remission [2]. RA used to be common in women in their 40s–50s; however, as the population ages, the number of patients with elderly onset RA (EORA), which occurs after 60 years of age, has been increasing [3]. Compared to young-onset RA (YORA), EORA patients are more likely to be male and seronegative for rheumatoid factor (RF) and anti-cyclic citrullinated protein (CCP) antibodies, and have markedly elevated C-reactive protein (CRP) levels and erythrocyte sedimentation rates, suggesting that EORA and YORA have a different pathophysiology [4,5,6]. In addition, patients with RA are at higher risk of developing malignancies, such as lymphomas, melanomas, and lung cancers, than healthy individuals [7,8]. Elderly patients with RA in particular are at high risk and often have a past or concurrent medical history of malignancy [9].

In recent years, immune checkpoint inhibitors (ICIs), such as anti-cytotoxic T-lymphocyte-associated protein 4 (CTLA-4), anti-programmed cell death protein-1 (PD-1), and anti-programmed cell death ligand-1 (PD-L1) antibodies, have emerged as new treatment options for malignancies [10,11,12]. These cancer immunotherapies activate T cells by inhibiting negative signals on T lymphocytes, thereby exerting an anti-tumor effect [13]. Initially tested for the treatment of malignant melanoma [14], these agents have been increasingly approved for a variety of cancer types, including renal, gastric, esophageal, colorectal, lung, and breast cancers [15]. Moreover, there are many ongoing clinical trials using checkpoint inhibitors that target novel costimulatory or coinhibitory molecules [16].

In parallel, there is increasing evidence that ICIs sometimes cause side effects called immune-related adverse events (irAEs), including hypophysitis, thyroiditis, myocarditis, and others [17]. Thus, irAEs cause diverse pathologies in nearly every organ [18,19]. Among them, conditions that resemble rheumatic diseases, such as arthritis, myositis, sicca symptoms, and vasculitis, are now termed rheumatic irAEs and are often treated with similar therapeutic agents as those for rheumatic diseases. However, rheumatic irAEs differ from idiopathic rheumatic diseases in multiple aspects, necessitating the elucidation of their pathophysiology.

This review summarizes the current evidence on the underlying mechanisms of irAEs. Then, the recent topics of rheumatic irAEs, particularly arthritis, myositis, and vasculitis, are updated. Finally, potential therapeutic strategies for rheumatic irAEs are discussed.

## 2. Immune-Related Adverse Events (irAEs)

### 2.1. Mechanisms of Action of Immune Checkpoint Inhibitors (ICIs)

Monoclonal antibodies targeting immunological checkpoints, currently termed ICIs, represent a growing class of agents effective against multiple tumor types and all disease stages [20]. Immune checkpoints are receptors expressed by immune cells that are dynamically involved in immune homeostasis and self-tolerance and particularly relevant to T cell function [21]. The development of ICIs has introduced a new era in cancer treatment, enabling the possible long-term survival of patients with metastatic disease and providing new therapeutic options in patients at an early stage. 

ICIs target different sets of regulatory interactions, the most common of which is the CTLA-4, PD-1, and PD-L1 blockade [15]. Upon T cell activation, multiple mechanisms initiate to control the level of activation. The two main regulatory interactions that control T cell activation are CTLA-4 (also called CD152) binding to CD80/86 (also called B7-1 and B7-2) on antigen-presenting cells and PD-1 binding to PD-L1 (also known as B7-H1) or programmed cell death ligand 2 (also known as B7-H2) [22]. Compared to PD-1, CTLA-4 has a more proximal role, acting as a higher-affinity competitor to CD28, thus limiting T cell activation at the priming stage [22]. These regulatory systems are targets for cancer immunotherapy, and the blockade of these immunoinhibitory receptors unleashes activated T cells and exerts anti-tumor effects. Combining anti-CTLA-4 antibodies with anti-PD-1 or anti-PD-L1 antibodies further enhances the anti-tumor effect [23].

However, ICIs are effective in only a small proportion of patients. Multiple attempts are ongoing to enhance their anti-tumor effects [24,25] or determine their efficacy at an early stage with biomarkers using peripheral blood, tissue samples, and imaging modalities [26,27,28,29,30,31]. In 2018, Matson et al. reported that the patients’ gut microbiome was associated with their response to ICIs [32]. However, the results differ among cohorts and their reproducibility has not been demonstrated [33].

### 2.2. Mechanisms of irAEs

In parallel with their success against malignancies, ICIs induce potentially severe and even lethal side effects called irAEs, which are considered off-target tissue damage and involve essentially every organ system [17]. A wide range of irAEs have been described [34,35], including hypophysitis, thyroiditis, myocarditis, pneumonia, pancreatitis, hepatitis, nephritis, adrenal insufficiency, enteritis, and skin rash (Figure 1). Although the underlying mechanisms of irAEs are not fully understood, (1) aberrant cytotoxic T cell activation, (2) increased autoantibody production, (3) direct molecular mimicry, (4) inflammatory cytokine production, (5) complement-mediated inflammation, (6) imparted regulatory T cell function, and other mechanisms contribute to the immunopathology of irAEs [17,19,36,37,38].

These irAEs can be provoked by both the CTLA-4 and PD-1 blockades, but their frequencies vary. For instance, colitis, hypophysitis, and rash occur more frequently with CTLA-4 inhibitors, whereas pneumonitis, hypothyroidism, and vitiligo are more common with PD-1/PD-L1 inhibitors [39,40,41]. Thus, the irAE profiles are distinct between the two groups. The molecular basis requires elucidation; however, recent studies demonstrated that anti-CTLA-4 and anti-PD-1 have different effects on T cell subsets [42]. CTLA-4 blockade primarily induces CD4^+^ T cell activation, particularly Th1 cells and follicular helper T cells, while counteracting regulatory T cell function [43,44,45,46]. PD-1 blockade mainly reinvigorates CD8^+^ T cells, including exhausted and cytotoxic T cells, while expanding CD4^+^ regulatory T cells [47,48,49]. These molecular differences may account for the clinical differences between the two.

Recent technical advances have provided interesting insights into irAEs. For example, myocarditis due to ICIs has long been reported [50]; however, using multiomics single-cell technology, Zhu et al. found an expansion of cytotoxic CD8^+^ T effector cells re-expressing CD45RA in patients with ICI-induced myocarditis [51]. These cells exhibit high levels of activation and cytotoxicity as well as chemokine receptor expression, suggesting that these cells have a pathogenic role and could be a therapeutic target in ICI-induced myocarditis [51]. In addition, genome-wide association studies have identified germline variants of *IL7* associated with all-grade irAEs [52,53]. Patients who carried the *IL7* germline variant showed increased lymphocyte stability after ICI initiation, high risk of all-grade irAEs, and improved survival [52,53]. As interleukin (IL)-7 is a critical regulator of T cell homeostasis [54,55], these results suggest that T lymphocytes are critically involved in the efficacy and safety of ICIs.

## 3. Rheumatic irAEs

### 3.1. Autoimmune-Disease-like Symptoms Due to ICIs

Since immunological checkpoints are critical in immune homeostasis and self-tolerance, the blockade of these molecules leads to immune system overactivation and the development of autoimmune-disease-like symptoms [56]. These are now called rheumatic irAEs and have been described in a growing number of case series and reports. The abrogation of immune checkpoints, such as CTLA-4 or PD-1, has similar consequences in animal models. The deletion of *CTLA4* in mice results in premature death due to lymphocyte proliferation and multiorgan failure [57,58]. In contrast, the deletion of *Pd1* or *Pdl1* results in less severe conditions but shows spontaneous arthritis and lupus-like glomerulonephritis [59,60]. Similar to irAEs, rheumatic irAEs are associated with diverse pathologies. The most common clinical manifestation is arthritis, followed by myositis, myalgia, vasculitis, Sjögren-syndrome-like sicca symptoms, a psoriasis-like rash, systemic-sclerosis-like skin sclerosis, and lupus-like nephritis (Figure 2).

Rheumatic irAEs resemble classical rheumatological diseases, suggesting a possible overlapping mechanism. Thus, elucidation of the disease mechanisms underlying rheumatic irAEs may lead to novel therapeutic approaches for rheumatic diseases [61]. Although the precise mechanism remains unclear, ICIs block negative signals on activated T cells, leading to increased autoreactive T cells and inflammatory cytokines such as tumor necrosis factor-α (TNF-α), IL-6, and IL-17 [62,63,64]. This provides the rationale for targeted inhibition of these cytokines to treat rheumatic irAEs [65,66]. ICIs can also induce epitope spreading, diversifying the epitope specificity of T cells [62]. Moreover, ICIs prevent the apoptosis of B cells, allowing for longer cell survival and increased autoantibody loads [62]. 

Rheumatic irAEs are often treated with therapeutic agents used for classical rheumatic diseases, such as glucocorticoids, conventional synthetic disease-modifying antirheumatic drugs (DMARDs), and biological DMARDs (bDMARDs) [67,68,69,70,71,72] (Figure 3). Among them, abatacept, a fusion protein of the extracellular domain of CTLA-4 and the Fc portion of immunoglobulin G, blocks the T-cell-activating interaction between CD28 and CD80/86 and is an effective bDMARD for RA [73,74]. Its mechanism is the conversion of the anti-CTLA-4 antibody, which blocks the T-cell-inhibitory interaction between CTLA-4 and CD80/86 [75,76]. For these reasons, abatacept should be avoided for the treatment of irAEs [77]; however, in cases of life-threatening conditions such as myocarditis, its use can be considered [78,79,80]. Hereafter, rheumatic irAEs will be discussed in each section. In particular, we will focus on arthritis, myositis, and vasculitis.

### 3.2. Arthritis

In recent years, reports of inflammatory arthritis after ICI administration have been increasing [81,82,83,84,85,86]. Arthritis is more common with PD-1/PD-L1 blockade than with CTLA-4 blockade [39], but the combination carries the highest risk [87]. In a retrospective cohort study, the prevalence of ICI-induced arthritis was estimated as 2% among patients undergoing cancer immunotherapy [87]. In this study, the mean age of 34 patients with ICI-induced arthritis was 59.2 years; 65% of them had the polyarticular type and 35% had the oligoarticular type, and the mean CRP level was 5.14 mg/dL. These characteristics were not distinguishable from RA. However, notably, RF and CCP antibodies were negative in all patients, suggesting that the pathophysiology differs between ICI-induced arthritis and RA and that autoantibodies are not directly involved in ICI-induced arthritis [87]. 

In contrast, according to a study by Cappelli, anti-RA33 antibodies were detected in 11.4% of patients with ICI-induced arthritis, whereas none of the ICI-treated controls without arthritis tested positive [88]. The antibodies were also present in 7.7% of patients with RA and 2% of healthy controls, suggesting that autoantibodies may play a role in ICI-induced arthritis or that individuals with pre-existing autoantibodies are at risk of ICI-induced arthritis.

According to the European Alliance of Associations for Rheumatology (EULAR), mild to moderate arthritis can be treated with nonsteroidal anti-inflammatory drugs, analgesics, and/or intra-articular glucocorticoids. If inflammation is evident despite these medications, systemic glucocorticoids should be considered [77]. Some may require high doses of glucocorticoids. However, we must consider that glucocorticoids may negatively impact the efficacy of cancer immunotherapy [89,90]. In animal models, both endogenous and exogenous glucocorticoids abolished anti-tumor immune responses [91]. Therefore, in patients with mild joint involvement, if feasible, systemic glucocorticoid administration should be avoided [90]. However, in the abovementioned retrospective study, 62% of patients were treated with systemic glucocorticoids [87], suggesting their high efficacy.

Conventional synthetic DMARDs, such as methotrexate, hydroxychloroquine, and sulfasalazine, and bDMARDs, such as TNF-α and IL-6 inhibitors, may be necessary in patients with moderate to severe arthritis that is refractory to glucocorticoid therapy [77]. Consultation with oncologists is necessary to decide whether to continue or stop ICI therapy and use bDMARDs. As mentioned above, the use of abatacept should be avoided because of the risk of antagonizing the anti-tumor effects of ICIs [77]. There is no evidence in the literature to suggest a preference for one biologic DMARD over another: infliximab, adalimumab, etanercept, tocilizumab, and sarilumab have controlled symptoms [56,66,92].

Some studies have suggested that this order of treatment should be reversed [56]. In other words, bDMARDs should be considered initially. The reasons for this are as follows: (1) ICI-induced arthritis also progresses to joint destruction from an early stage [93]; and (2) bDMARDs may not interfere with the anti-tumor response of ICI [82,94,95]. For instance, an in vitro study compared the effects of dexamethasone and infliximab on the anti-tumor activity of tumor-infiltrating lymphocytes. Even low doses of dexamethasone significantly attenuated anti-tumor activity, whereas a standard dose of infliximab had only minor effects on tumor killing. The activity of tumor-infiltrating lymphocytes was restored after the discontinuation of dexamethasone [96]. Therefore, bDMARDs have been administered in an increasing number of cases of ICI-induced arthritis [97].

A recent molecular study demonstrated that CD8^+^ T cells from ICI-treated patients who developed arthritis had distinct effector functions and metabolic profiles from those from ICI-treated patients who remained arthritis-free and that TNF-α inhibitors or Janus kinase (JAK) inhibitors were unable to adequately correct such dysfunction in vitro [98]. This suggests that bDMARDs and JAK inhibitors may effectively reduce symptoms but may not fundamentally restore T cell function, although because of the risk of malignancy [99], JAK inhibitors are normally avoided in ICI-induced arthritis.

We have so far discussed newly onset arthritis or “de novo” arthritis after ICI initiation, but what happens when ICIs are given in patients with pre-existing arthritis? Most of the clinical trials have excluded patients with pre-existing autoimmune disease because of the concern of disease flare; however, real world evidence is accumulating that ICIs can be administered to such patients with an acceptable safety profile, but the likelihood of disease flares is high [87,100,101]. For instance, the relapse rates for RA and psoriatic arthritis have been reported to be greater than 50%, while those for systemic lupus erythematosus, systemic sclerosis, myositis, and vasculitis range from 20 to 50%, indicating that disease flare varies depending on the type of pre-existing autoimmune disease. Even when RA flares, symptoms are relatively mild and ICIs can often be resumed [100]. 

More recent evidence has shown that patients with pre-existing autoimmune disease are at greater risk of all-grade, severe, and multiple irAEs, but have a better survival than controls [102]. These results indicate that having an autoimmune disease is not a contraindication to ICI and that T cells from patients with autoimmune disease may be more susceptible to ICI treatment than those from patients without autoimmune disease.

### 3.3. Myositis

Compared to arthritis, ICI-induced myositis is rare, with an estimated incidence of less than 1% but a high mortality rate of up to 22% [87,103,104,105]. The meta-analysis demonstrated no significant difference in the incidence of anti-PD-1/PD-L1 and anti-CTLA-4 antibodies (odds ratio, 1.07; 95% confidence interval, 0.27–9.41) [106]; however, anti-CTLA-4 antibodies have been reported less frequently than anti-PD-1/PD-L1 antibody treatment. The average age at onset of myositis is approximately 70 years [87,106,107], and the mean time from ICI initiation to myositis onset is approximately 4 weeks [106,107]. ICI-induced arthritis often occurs several months after the initiation of ICI [29], whereas myositis often develops at a relatively early treatment phase.

Classical inflammatory myositis includes dermatomyositis (DM) and polymyositis (PM), and ICI-induced myositis is indistinguishable from these in many aspects. Creatine kinase (CK) levels often increase to >1000 units/L, electromyography shows myogenic changes, and high-intensity signals can be detected in affected lesions on T2-weighted magnetic resonance imaging [87,108]. A skin rash characteristic of DM is found in 20% of patients [106]. However, ICI-induced myositis includes clinical features that are distinct from those of idiopathic inflammatory myositis. Unlike PM/DM, ICI-induced myositis often presents with ptosis and ocular motility disorders as initial symptoms [106]. Interstitial lung disease is detected in only a small proportion of patients, approximately 2%, while up to 40% of patients develop myocarditis. Respiratory muscle involvement is also common in ICI-induced myositis. Ptosis and respiratory muscle involvement are predictive of myocarditis development [106].

Most cases of ICI-induced myositis test negative for myositis-specific/-associated antibodies [106], whereas some patients show multiple myositis-specific/-associated antibodies [109]. Moreover, anti-striated muscle antibodies have often been detected in ICI-induced myositis. Anti-striated muscle antibodies include anti-titin antibodies and anti-Kv1.4 antibodies, and are often found in patients with thymoma-associated myasthenia gravis [110], suggesting an overlap in the pathophysiology between ICI-induced myositis and myasthenia gravis [87]. Thus, a fatal triad of myasthenia gravis, myositis, and myocarditis can occur in patients with ICI-induced myositis [111]. However, marked improvement in the edrophonium test, which is observed in myasthenia gravis, is rare in ICI-induced myositis [107].

Muscle biopsy shows myofiber necrosis with regenerative findings and infiltration of CD4^+^ T cells, CD8^+^ T cells, CD20^+^ B cells, and CD68^+^ macrophages. High expression of major histocompatibility complex class I is also observed. These findings are slightly different from those in anti-signal recognition particles and anti-aminoacyl-tRNA synthetase antibody-positive myositis [107]. Human leucocyte antigen (HLA) class I haplotypes (HLA-A*24:02/B*52:01/C*12:02) can be frequently detected in ICI-induced myositis [107].

According to the clinical practice guidelines for the management of irAEs [112], mild to moderate myositis can be treated with prednisolone from 10 mg to 1 mg/kg, depending on severity. In severe cases, it is recommended to start with prednisolone 1 mg/kg and consider methylprednisolone 1–2 mg/kg or a high-dose bolus. Additionally, plasma exchange therapy and/or intravenous immunoglobulin therapy should be considered [77]. If symptoms and CK levels do not improve or worsen despite treatment, other immunosuppressive agents, such as methotrexate, azathioprine, or mycophenolate mofetil, should be considered [112]. Resumption of ICIs is usually avoided when myositis, particularly in association with myocarditis, develops, although there are some reports with successful resumption of ICIs in patients with mild disease [107].

### 3.4. Vasculitis

Cancer immunotherapy is a potential trigger for vasculitis [72,113,114,115]. Vasculitis is classified into large-, medium-, and small-vessel types depending on blood vessel size [116], but ICIs can cause any type of vasculitis. In fact, giant cell arteritis (GCA) [117,118], aortitis and large-vessel involvement [119,120,121], IgA vasculitis [122,123], granulomatosis with polyangiitis [124,125], eosinophilic granulomatosis with polyangiitis [126], and cutaneous small-vessel vasculitis [127,128] have been reported after ICI administration. A retrospective pharmacovigilance study evaluated the association between ICIs and cardiovascular adverse events and demonstrated that ICI treatment was associated with a higher risk of vasculitis when compared with the database obtained from individual case safety reports [129]. This signal was primarily driven by GCA, with a reporting odds ratio of 12.99. Anti-PD-1/PD-L1 antibodies have a relatively higher risk of developing vasculitis than anti-CTLA-4 antibodies, but the combination of these have the highest risk [129].

Although the mechanisms underlying the development of GCA by ICIs are unclear, we have previously demonstrated that blockade of PD-1/PD-L1 signaling exacerbated vascular inflammation using a preclinical mouse model of large-vessel vasculitis [13,130]. In this mouse model, human medium- and large-sized arteries are embedded into NOD scid immunodeficient mouse, and then peripheral blood mononuclear cells from patients with GCA are transferred into this mouse, causing vasculitis by alloreaction [131]. The administration of anti-PD-1 antibody into this mouse not only increased the infiltration of activated T cells in the vasculature and inflammatory cytokines, such as IL-6, IL-17, TNF-α, and interferon gamma (IFN-γ), but also promoted hyperplasia in the intima and neovessel formation in the adventitia, suggesting that the PD-1/PD-L1 signal is critically involved in vascular remodeling [130]. 

These experiments prompted us to examine the expression of PD-L1 on antigen-presenting cells in the vasculature. We found that PD-L1 expression was markedly diminished in dendritic cells (DCs) within the adventitia. These results suggest that deficient PD-L1 expression on vascular DCs was unable to sufficiently inhibit T cell overactivation in vascular lesions and that ICI administration mimics GCA immunopathology [132]. Reduced expression of PD-L1 in vascular DCs has been confirmed in a mouse model of lipopolysaccharide-induced vasculitis [133]. Moreover, aberrant PD-1/PD-L1 signal expression has also been reported in anti-neutrophil cytoplasmic-antibody-associated vasculitis [134,135]. Although a variety of drugs reportedly cause drug-induced vasculitis, such as hydralazine, minocycline, propylthiouracil, granulocyte colony-stimulating factor, and anti-TNF-α inhibitors [136,137], it is important to recognize that ICIs also can trigger any type of vasculitis.

Vasculitis caused by ICIs usually requires high-dose glucocorticoid treatment [77]. Other therapeutic options include cyclophosphamide [138], tocilizumab [119], rituximab [139,140], plasmapheresis [141], and JAK inhibitors [142,143]; however, the optimal management of ICI-induced vasculitis has not been established.

### 3.5. Myalgia (Polymyalgia Rheumatica)

Polymyalgia rheumatica (PMR) is the most common inflammatory rheumatic disease affecting individuals >50 years of age [144]. PMR is characterized by pain in the shoulder and pelvic girdles and often associated with GCA [145]. Many patients respond quickly to low-dose glucocorticoids; however, multiple relapses are common with glucocorticoid tapering [146,147].

PMR recently emerged as a side effect of cancer immunotherapy [148,149,150,151,152,153,154]. Compared with anti-CTLA-4 antibodies, anti-PD-1/PD-L1 antibodies have a higher risk of inducing this manifestation [103]. Ultrasound and 18F-fluorodeoxyglucose-positron emission tomography (FDG-PET) findings are comparable to those of idiopathic PMR [154]. Most patients fulfill the classification criteria for PMR [149,155] and respond well to low-dose glucocorticoids [149]. If patients respond inadequately to glucocorticoids, conventional synthetic DMARDs, such as methotrexate, hydroxychloroquine, and sulfasalazine, and bDMARDs, such as tocilizumab, may be effective [69,149,156].

### 3.6. Sicca Symptoms (Sjögren Syndrome)

Salivary gland dysfunction is a hallmark of Sjögren syndrome [157]. ICI-induced sicca syndrome (ICIS), characterized by severe sudden-onset dry mouth [158], has been increasingly reported [82,159,160,161]. Patients with anti-PD-1/PD-L1 antibodies are at a relatively higher risk of developing this symptom than those with anti-CTLA-4 antibody [103]. Pathologically, the presence of focal lymphocytic sialadenitis is a shared feature of Sjögren syndrome and ICIS, but the nature of immune cell infiltration is T-cell-dominant in ICIS but B-cell-dominant in Sjögren syndrome [162]. Pringle et al. proposed that ICIS is inflammation-driven by IFN-γ and represents a new type II interferonopathy [162]. In many cases, low-dose glucocorticoids improve subjective symptoms but salivary gland function does not recover [161].

### 3.7. Skin Rash (Psoriasis)

Psoriasis-like skin rash can be observed as a rheumatic irAE after ICI administration [163,164,165,166,167]. A retrospective study showed that among 8683 ICI-treated patients, 262 (3%) developed psoriasis-like cutaneous irAEs [163]. Of these, 5% of patients with ICI-induced psoriasis showed inverse psoriasis, which required differentiation from fungal infection [163]. Flares of pre-existing psoriasis and psoriatic arthritis are also frequent, with some reports of more than 50% [101,168,169]. 

### 3.8. Skin Sclerosis (Systemic Sclerosis)

Systemic-sclerosis-like skin sclerosis or skin thickening after ICI use was initially thought to be an extremely rare rheumatic irAE [170], but a recent review suggests that it may be more common than ever thought [171]. Anti-PD-1 antibodies are at higher risk of skin sclerosis than anti-CTLA-4 antibodies, and renal crisis can be observed at the onset [171].

### 3.9. Lupus-like Disease

Other rheumatic irAEs include lupus-like disease [172,173,174]. Not only skin lesions but also nephritis can be observed [174,175]. The types of rheumatic irAEs may increase as ICI usage increases.

## 4. A Potential Strategy to Identify Patients at Risk of Rheumatic irAEs

How can we identify patients at risk of developing rheumatic irAEs? It has been reported that ANA and autoantibodies are prevalent in patients with rheumatic irAEs [176]; however, whether ANA and autoantibodies should be examined in all patients before starting ICIs is debatable. According to the EULAR points to consider for the diagnosis and management of rheumatic irAEs, there is no indication to test all patients for the presence of autoantibodies [77]. In order to identify patients at risk of rheumatic irAEs, a comprehensive framework for analyzing HLA, ANA, autoantibodies, cytokines and complement levels, and cytotoxic and regulatory T cell function may be required. Although ANA immunofluorescence pattern helps identify autoimmune liver disease [177,178], it remains unclear whether ANA patterns are predictive of rheumatic irAEs.

## 5. Optimal Management of Rheumatic irAEs

As we have seen so far, ICIs cause a variety of rheumatic irAEs that may further increase (Table 1). A multidisciplinary discussion between rheumatologists, oncologists, and patients is of utmost importance for the optimal management of rheumatic irAEs [77]. The continuation or discontinuation of ICIs should be guided by their effectiveness, the severity of irAEs, and the intensity of immunosuppressive therapy. In addition, therapeutic agents not only need to control symptoms, but also maintain the anti-tumor effects of ICIs [77]. Despite growing interest, prospective trials have not yet been conducted, and evidence is still lacking regarding optimal management that also permits the anti-tumor effects of ICIs. In animal models, the prophylactic administration of TNF-α inhibitors or IL-6 inhibitors maintained ICI efficacy with minor toxicity [179,180]. However, these therapeutic strategies have not been proven in humans.

## 6. Conclusions

A multidisciplinary and interdisciplinary approach is critical to facilitate the early diagnosis and treatment of rheumatic irAEs to prevent permanent organ damage. Further studies are required to determine their optimal management.

## Figures and Tables

**Figure 1 ijms-24-05643-f001:**
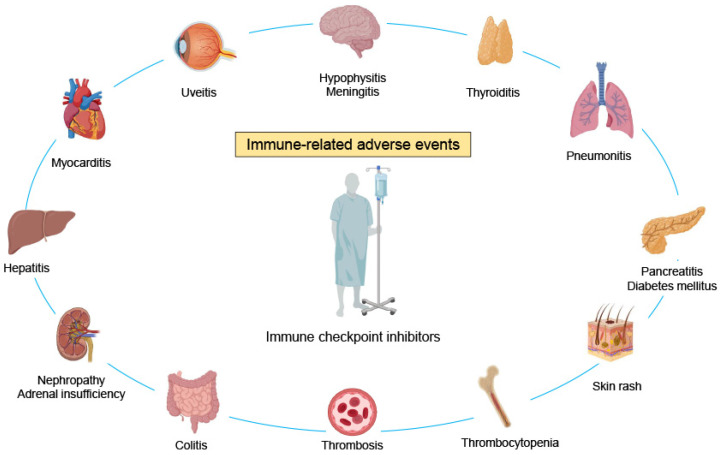
Immune-related adverse events affect multiple organs. Immune checkpoint inhibitors can cause side effects called immune-related adverse events (irAEs), which involve hypophysitis, meningitis, thyroiditis, uveitis, pneumonitis, myocarditis, pancreatitis (diabetes mellites), hepatitis, erythema, nephropathy, adrenal insufficiency, thrombocytopenia, colitis, thrombosis, and other manifestations. Colitis, hypophysitis, and skin rash occur more frequently with anti-cytotoxic T-lymphocyte-associated protein 4 (CTLA-4) antibodies, whereas pneumonitis, hypothyroidism, and vitiligo are more common with anti-programmed cell death protein-1 (PD-1) and anti-programmed cell death ligand-1 (PD-L1) antibodies.

**Figure 2 ijms-24-05643-f002:**
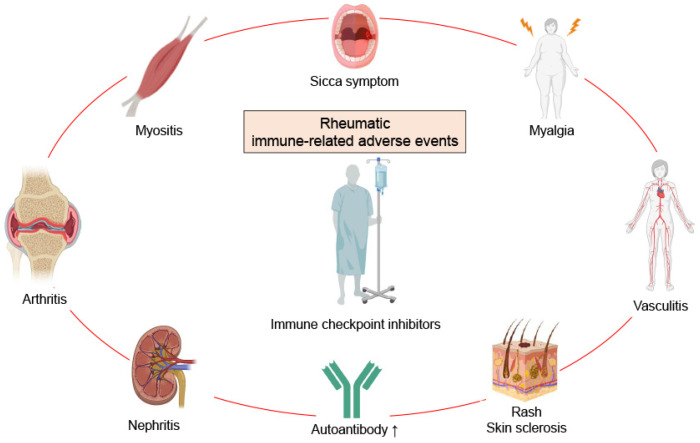
Rheumatic immune-related adverse events show diverse pathologies. Immune checkpoint inhibitors sometimes cause rheumatic-disease-like symptoms called rheumatic immune-related adverse events (irAEs), which involve arthritis, myositis, myalgia, sicca symptoms, a rash, skin sclerosis, nephritis, and vasculitis. Autoantibody production may also be increased. Rheumatic irAEs more commonly occur with anti-programmed cell death protein-1 (PD-1) and anti-programmed cell death ligand-1 (PD-L1) antibodies than anti-cytotoxic T-lymphocyte-associated protein 4 (CTLA-4) antibodies.

**Figure 3 ijms-24-05643-f003:**
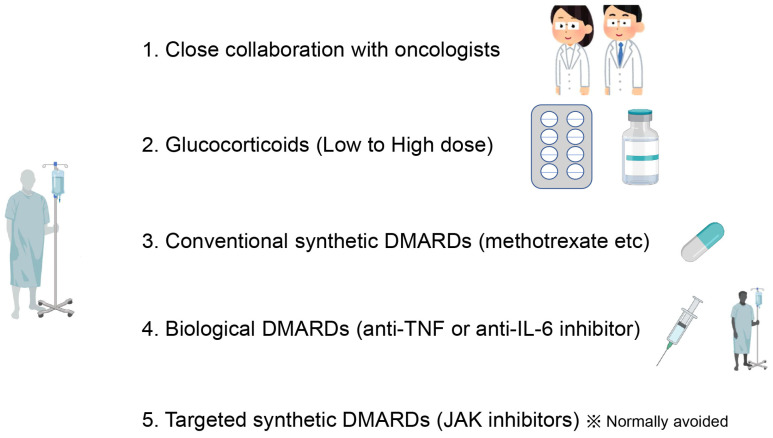
General management of rheumatic immune-related adverse events (irAEs) due to cancer immunotherapy. Consultation with oncologists is required to decide whether to continue or terminate immune checkpoint inhibitors. If patients are refractory to nonsteroidal anti-inflammatory drugs, systemic glucocorticoids should be considered. Based on the severity, conventional synthetic disease-modifying antirheumatic drugs (DMARDs) and biological DMARDs, such as anti-tumor necrosis factor (TNF) or interleukin (IL)-6 inhibitor, should be considered. Targeted synthetic DMARDs, such as Janus kinase (JAK) inhibitors, are normally avoided because of the risk of exacerbating malignancies.

**Table 1 ijms-24-05643-t001:** Brief summary of rheumatic immune-related adverse events (irAEs).

	Incidence	Treatment
Arthritis	Up to 5%	1. NSAIDs, 2. Glucocorticoids, 3. DMARDs
Myositis	<1%	1. Glucocorticoids, 2. DMARDs, 3. PE, 4. IVIg
Vasculitis	<1%	1. Glucocorticoids, 2. CYC, 3. RTX, 4. TCZ
Myalgia	Up to 5%	1. Glucocorticoids, 2. DMARDs, 3. TCZ
Sicca symptom	Rare	1. Glucocorticoids
Skin rash (psoriasis)	Up to 3%	1. Glucocorticoids, 2. DMARDs
Skin sclerosis	Rare	1. Glucocorticoids, 2. CYC, 3. DMARDs
Lupus-like disease	Rare	1. Glucocorticoids, 2. CYC

Treatment should be decided based on the severity. CYC: cyclophosphamide, DMARDs: disease-modifying antirheumatic drugs, IVIg: intravenous immunoglobulin, NSAIDs: nonsteroidal anti-inflammatory drugs, PE: plasma exchange, RTX: rituximab, TCZ: tocilizumab.

## Data Availability

Not applicable.

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
