# Peer review of "Rheumatic Immune-Related Adverse Events due to Immune Checkpoint Inhibitors—A 2023 Update"

_ijms, 2023, doi:10.3390/ijms24065643_

Round 1

Reviewer 1 Report

In this review article, the authors summarized current available data on the immune-related adverse events (irAEs) developing in patients treated with immune checkpoint inhibitors (ICIs). They also discussed potential therapeutic strategies for rheumatic irAEs.

The review is well written and address a topic of current major clinical interest. However, some issues should also discussed and addressed. 

-Mechanisms of irAEs: it has been recently suggested that a prominent role in irAEs pathogenesis is related to the impaired function of CD4+ CD25+ Foxp3 regulatory T cells (Treg), a subset of T cell population with immunosuppressive role that express a panel of chemokine receptors and surface molecules such as CTLA4, PD-1 and others, thus potentially making them a very direct target of ICI immunotherapy, as recently described (Hepatocellular carcinoma in viral and autoimmune liver diseases: Role of CD4+ CD25+ Foxp3+ regulatory T cells in the immune microenvironment. World J Gastroenterol. 2021 Jun 14;27(22):2994-3009.).

-In my opinion, it would be of clinical relevance discuss also the diagnostic role of autoantibodies that should be searched in all patients before starting ICIs since they could suggest a pre-existing underlying autoimmunity that could be triggered by ICIs. In this regard, however, the detection of antinuclear antibodies (ANA) may have a diagnostic significance according to the different ANA immunofluorescence pattern since ANA of "rheumatologic" significance are different from ANA associated to autoimmune liver diseases, as previously demonstrated (DOI: 10.1111/j.1365-2036.2006.03172.x; doi: 10.1038/s41584-021-00573-7.).

-A paragraph addressing a potential strategy to identify patients at higher risk of rheumatic irAEs (autoantibody screening? HLA associated to rheumatic/autoimmune diseases? other?) would further improve the manuscript.

(Fig

Author Response

Thank you very much for your comments. We appreciate your time for reviewing our manuscript.

Comments form the reviewer

-Mechanisms of irAEs: it has been recently suggested that a prominent role in irAEs pathogenesis is related to the impaired function of CD4+ CD25+ Foxp3 regulatory T cells (Treg), a subset of T cell population with immunosuppressive role that express a panel of chemokine receptors and surface molecules such as CTLA4, PD-1 and others, thus potentially making them a very direct target of ICI immunotherapy, as recently described (Hepatocellular carcinoma in viral and autoimmune liver diseases: Role of CD4+ CD25+ Foxp3+ regulatory T cells in the immune microenvironment. World J Gastroenterol. 2021 Jun 14;27(22):2994-3009.).

Response: Thank you very much for your excellent comments. Based on your comments, we have cited this paper in the manuscript (Line 101, reference 36).

-In my opinion, it would be of clinical relevance discuss also the diagnostic role of autoantibodies that should be searched in all patients before starting ICIs since they could suggest a pre-existing underlying autoimmunity that could be triggered by ICIs. In this regard, however, the detection of antinuclear antibodies (ANA) may have a diagnostic significance according to the different ANA immunofluorescence pattern since ANA of "rheumatologic" significance are different from ANA associated to autoimmune liver diseases, as previously demonstrated (DOI: 10.1111/j.1365-2036.2006.03172.x; doi: 10.1038/s41584-021-00573-7.).

Response: Thank you very much for your excellent comments. We agree with the reviewer. Based on your comments, we have cited these papers in the manuscript (Line 389-398, reference 177, 178).

-A paragraph addressing a potential strategy to identify patients at higher risk of rheumatic irAEs (autoantibody screening? HLA associated to rheumatic/autoimmune diseases? other?) would further improve the manuscript.

Response: Thank you very much for your excellent comments. We totally agree with the reviewer. Based on your comments, we have added the paragraph (Line 388-398).

Reviewer 2 Report

“Rheumatic immune-related adverse events due to immune 2 checkpoint inhibitors-A 2023 update” is an excellent review of the immune-related adverse events due to immunotherapy in patients with cancer. The update is well written and structured.

I only have a few minor comments:

Since in recent works on JAK inhibitors (PMID: 35081280) there is some uncertainty as to whether or not they cause cancer, the indication for treatment of irAEs in patients receiving ICIs with JaK inhibitors seem something questionable, is it not? The authors should clarify this aspect.

I would suggest to the authors to include in the introduction or discussion an exhaustive and recent review on this subject from 102 cases in a real-world setting performed in Spain (PMID: 32896258).

In order to improve the perspective of the reader on the topic, I also suggest including a general figure or algorithm on the authors' recommendation for the management of irAEs in patients undergoing treatment with ICIs.

Author Response

Thank you very much for your comments. We appreciate your time for reviewing our manuscript.

Comments form the reviewer

Since in recent works on JAK inhibitors (PMID: 35081280) there is some uncertainty as to whether or not they cause cancer, the indication for treatment of irAEs in patients receiving ICIs with JaK inhibitors seem something questionable, is it not? The authors should clarify this aspect.

Response: Thank you very much for your comments. We total agree with the reviewer. Based on your comments, we have revised the manuscript (Line 2329-242).

I would suggest to the authors to include in the introduction or discussion an exhaustive and recent review on this subject from 102 cases in a real-world setting performed in Spain (PMID: 32896258).

Response: Thank you very much for your comments. We have cited this paper (Line 97, Reference 34).

In order to improve the perspective of the reader on the topic, I also suggest including a general figure or algorithm on the authors' recommendation for the management of irAEs in patients undergoing treatment with ICIs.

Response: Thank you very much for your comments. Based on your comments, we newly made Figure 3 and explained general management of rheumatic irAEs.

Reviewer 3 Report

In this review ,, Rheumatic immune-related adverse events due to immune 2 checkpoint inhibitors-A 2023 update-,, by Quang Minh Danget al. , the authors aimed to summarize:

i) the current evidence on the underlying mechanisms of the immune-related adverse events (irAEs), including hypophysitis, thyroiditis, myocarditis, and others;

ii) the recent topics of rheumatic irAEs, particularly arthritis, myositis, and vasculitis, are updated;

iii) finally, potential therapeutic strategies for rheumatic irAEs are discussed.

- The review is well structured, organized

- Through the analyzed topic and the mentioned recent studies, the work makes a significant contribution to the field

- Most of the bibliographic references are recent, even very recent, and adequate to related and previous work

- The paper is scientifically sound and not misleading

Author Response

Thank you very much for your comments. We appreciate your time for reviewing our manuscript.

Reviewer 4 Report

The Adverse immune events to checkpoints inhibitors remains an not enough explored area.

I consider important the topic, but I have several sugestions for authors :

please put in a table the frequencies of each subgroup of  adverse reactions/  pye

table with  each manifestations and the appropriate treatment(not only narative description), with percentages of response rates.

Putting the data more clearlly, the utility of the review will be highlithed. 

According to the European Alliance of Associations for Rheumatology (EULAR), please review the name .European League Against Rheumatology 197

Author Response

Thank you very much for your comments. We appreciate your time for reviewing our manuscript.

please put in a table the frequencies of each subgroup of adverse reactions/ pye 

table with  each manifestations and the appropriate treatment (not only narrative description), with percentages of response rates.

Putting the data more clearly, the utility of the review will be highlithed. 

Response: Thank you very much for your excellent comments. We totally agree with the reviewer. Therefore, we made Table 1. But it was extremely difficult to summarize the response rate for each treatment because of the insufficient evidence.

According to the European Alliance of Associations for Rheumatology (EULAR), please review the name .European League Against Rheumatology 197

Response: Thank you very much for pointing it out. According to the website (https://www.eular.org/), EULAR was formerly called the European League Against Rheumatism, but currently called the European Alliance of Associations for Rheumatology (EULAR). Should we revise the manuscript?

In addition, before submitting the manuscript, we have asked Editage to edit language and grammar accuracy. Certificate is attached.

Round 2

Reviewer 1 Report

In the revised manuscript all raised points have been addressed.